

# Maple and hickory leaf litter fungal communities reflect pre-senescent leaf communities

Julian A. Liber[1,2], Douglas H. Minier[3], Anna Stouffer-Hopkins[3],
Judson Van Wyk[4], Reid Longley[4] and Gregory Bonito[3,4]

[1] Department of Plant Biology, Michigan State University, East Lansing, Michigan, United States
[2] Department of Biology, Duke University, Durham, North Carolina, United States
[3] Department of Plant, Soil, and Microbial Sciences, Michigan State University, East Lansing, Michigan, United States
[4] Department of Microbiology and Molecular Genetics, Michigan State University, East Lansing, Michigan, United States

Corresponding author
Julian A. Liber, julian.liber@duke.edu

## ABSTRACT

Fungal communities are known to contribute to the functioning of living plant microbiomes as well as to the decay of dead plant material and affect vital ecosystem services, such as pathogen resistance and nutrient cycling. Yet, factors that drive structure and function of phyllosphere mycobiomes and their fate in leaf litter are often ignored. We sought to determine the factors contributing to the composition of communities in temperate forest substrates, with culture-independent amplicon sequencing of fungal communities of pre-senescent leaf surfaces, internal tissues, leaf litter, underlying humus soil of co-occurring red maple (*Acer rubrum*) and shagbark hickory (*Carya ovata*). Paired samples were taken at five sites within a temperate forest in southern Michigan, USA. Fungal communities were differentiable based on substrate, host species, and site, as well as all two-way and three-way interactions of these variables. PERMANOVA analyses and co-occurrence of taxa indicate that soil communities are unique from both phyllosphere and leaf litter communities. Correspondence of endophyte, epiphyte, and litter communities suggests dispersal plays an important role in structuring fungal communities. Future work will be needed to assess how this dispersal changes microbial community functioning in these niches.

## INTRODUCTION

Plant and soil microbiomes contribute to critical ecosystem functions and are involved in complex interactions within communities (*Delgado-Baquerizo et al., 2016*; *Wagg et al., 2019*; *Regalado et al., 2020*). The community of fungi present in the plant microbiome can colonize healthy roots, leaves, stems, and seeds (*Porras-Alfaro & Bayman, 2011*; *Floc'h et al., 2020*). Additionally, fungal communities have been implicated in disease susceptibility (*Gu et al., 2020*), nutrient acquisition and cycling (*Herzog et al., 2019*), and stress tolerance or resilience (*Waller et al., 2005*; *Márquez et al., 2007*). Yet, fungi also

occupy several critical roles in the plant and soil microbial communities, functioning as plant pathogens (*Brader et al., 2017*), hyperparasites (*Falk et al., 1995*; *Vandermeer, Perfecto & Liere, 2009*), mycorrhizal associates (*Soudzilovskaia et al., 2019*), saprobes (*Zhou & Hyde, 2001*), and specialized litter (*Osono, 2007*) or wood degraders (*Schilling et al., 2020*).

Studying the distribution of fungal communities within environments and between living and non-living substrates in a common environment may elucidate the natural history and functional ecology of these organisms (*Peay, 2014*). Priority effects (*Fukami et al., 2010*) and environmental filters, such as leaf traits, may be expected to have a role in the assembly of these communities (*Purahong et al., 2016*). The occurrence of a given taxon across multiple samples suggests dispersal (*Bell, 2010*). Dispersal processes can lead to priority effects if early arrivals establish and alter the assembly of the community (*Hiscox et al., 2015*). Furthermore, if environmental filters drive fungal community assembly, samples experiencing similar environments will be expected to be more similar in composition compared to those from environments that differ (*Keddy, 1992*). The processes of dispersal and filtering are not mutually exclusive, but instead may vary in importance depending on scale, taxa examined, and other factors of the study system (*Kivlin et al., 2014*; *Evans, Martiny & Allison, 2017*).

Fungal endophytes are defined as living asymptomatically within plant tissues (*Arnold et al., 2000*). The diversity and composition of endophytes can be distinguished from epiphytic communities, the microbes observed on surfaces of plant tissues, but some overlapping taxa are common (*Gomes et al., 2018*; *Yao et al., 2019*). In fact, epiphyte communities are presumed to give rise to endophyte communities, as they penetrate openings, such as stomata or damaged tissues (*Porras-Alfaro & Bayman, 2011*). Some authors have hypothesized potential ecological roles of endophytic fungi, including as plant pathogens, latent saprotrophs, or mutualists (*Veneault-Fourrey & Martin, 2011*; *Brader et al., 2017*; *Chen et al., 2018*). Following senescence of above-ground tissues, some of the endophytes and epiphytes (collectively the phyllosphere community) may switch nutritional modes to become active saprotrophs (*Promputtha et al., 2007*; *He et al., 2012*; *Gundel et al., 2017*) in forest floor leaf litter communities.

Examinations of plant-associated fungal communities have implicated multiple drivers in community composition and diversity. Rhizosphere communities have shown to be structured in-part by plant host species, location, and land-use effects (*Bonito et al., 2019*; *Schöps et al., 2020*). Similarly, aboveground tree endophytes showed significant discrimination based on plant tissue, host species, and site in a hemiboreal forest (*Küngas, Bahram & Põldmaa, 2020*). Host species identity may account for variation in endophyte community composition even while phylogenetic distance does not (*Kembel & Mueller, 2014*; *Whitaker et al., 2020*), which may instead be accounted for by functional traits (*Kembel & Mueller, 2014*). Variation in leaf secondary metabolites has also been linked with fungal endophyte community composition in some cases (*Christian et al., 2020*). In a more detailed examination of site effects, environmental characteristics such as rainfall and elevation were demonstrated as important factors in fungal endophyte community composition across the range of a genetically undifferentiated host tree population

(*Zimmerman & Vitousek, 2012*). These site effects may be more pronounced than genetic effects, as was shown in ecotypes of switchgrass (*Panicum virgatum*) grown in local and remote common gardens (*Whitaker, Reynolds & Clay, 2018*).

In the current study, we ask whether endophytic and epiphytic fungal phyllosphere communities of forest tree species differ, and whether phyllosphere fungal taxa may persist in leaf litter or soil communities. To address these questions, fungal communities of pre-senescent leaves, leaf litter and soils were assessed with high-throughput amplicon sequencing and compared. We compared fungal communities of red maple (*Acer rubrum* L.) and shagbark hickory (*Carya ovata* (Mill.) K.Koch) host tree species at five sites within a forest ecosystem in southern Michigan where the species co-occur. Based on previous studies of fungal endophytes, we hypothesized that phyllosphere fungal communities between the two plant species would differ. We also expected that many detected endophytic taxa would persist in leaf litter samples, but less so in soils. To determine how these niches structured communities, we sampled across four forest substrates: (1) the surface of pre-senescent leaves, (2) the internal tissues of pre-senescent leaves, (3) leaf litter, and (4) the soil in direct contact with the leaf litter.

Sampling protocols and downstream sample processing can have overlooked effects on the microbial community analyses (*Hallmaier-Wacker et al., 2018*). The methods and materials chosen can bias the community observed through DNA extraction and PCR/primer decisions (*Brooks et al., 2015*). While much research has focused on comparison of kit or DNA extraction methods (*Vo & Jedlicka, 2014*; *Brooks et al., 2015*; *Hallmaier-Wacker et al., 2018*; *Angebault et al., 2020*), we did not find literature assessing swab material type for observation of phyllosphere communities. A second aim of this study was to compare the efficacy of swab types composed of two different materials in assessing epiphytic leaf communities with the goal of promoting cost-efficient yet consistent and thorough environmental sampling.

## MATERIALS AND METHODS

### Site

Samples were collected within a 9-hectare quadrat in a semi-homogeneous deciduous forest site within Dansville State Game Area, Dansville, Michigan, USA (N 42.5171, W 84.3260, altitude 291 m) and approved by the Michigan Department of Natural Resources. Soil types in the quadrat included loam, sandy loam, and loamy sand, and fall under the following USDA soil taxonomy classifications: Oxyaquic Glossudalfs, Typic Endoaquolls, Aquollic Hapludalfs, Aquic Arenic Hapludalfs (Fig. S1). Mean annual climate observed at Jackson County Airport, 30.7 km SSW of the site, was 800 mm precipitation, 9.1 °C temperature, and 69% relative humidity. Altitude measurements were performed using Google Earth Pro (ver. 7.3.2). Distances between sites were determined using the "Measure distance" function in Google Maps.

### Sample collection

Samples were collected on September 16, 2018 from five sites 76–246 m apart (Table S1) selected within the study area described above. These include one site each in the

northeast, northwest, southeast, southwest, and center areas on the quadrat. At each site at least one shagbark hickory tree (*Carya ovata*), one red maple tree (*Acer rubrum*) and at least 10 leaves or leaflets of each host's leaf litter beneath the canopy of sampled trees were present. We selected these tree species as they were consistently found across the forest, providing the opportunity to test site effects. The litter collected was intact leaves of the target host, that were presumed to have fallen the previous year. Ten fresh leaves (*Acer*) or leaflets (*Carya*) per tree per site were collected axenically from branches within reaching or jumping height (<3 m from ground) and swabbed in the field as a single pooled sample per site, host tree species, and swab type, then transported to the laboratory for further processing. The entire surface of the leaf or leaflet, both top and bottom, was swabbed for those used. For each site and tree, three leaves or leaflets of litter were collected (separately for each host species), and the topsoil (mull humus, about 10 g) immediately below the leaves was collected with a sterilized metal scoop. One pooled sample was produced for each combination of site, host tree species, and substrate for the leaf endophyte, litter, and soil samples. Because only a single tree for each species at each site was sampled, we were not able to differentiate whether inter-individual variation was due to within-species genotype variability or site environment variability. All collected leaf and soil materials were stored in sterile sealed plastic bags, placed in an insulated cooler, brought directly back to the lab, and stored at 4 °C until processing (over the following week). All samples of a given substrate were processed on the same day.

Ten leaves or leaflets of each tree at each site were swabbed with two swabs that were dipped in Extraction Solution (ES-100 mM Tris, 250 mM KCl, 10 mM disodium EDTA, adjusted to pH 9.5–10) directly prior to sampling, producing two extractions per tree and site. Leaves were swabbed with both cotton-tipped applicators and polymer-tipped PurFlock ULTRA® applicators (Puritan Medical Products, Guilford, ME, USA) referred to as "cotton" and "synthetic", respectively. Swab heads were broken off into individual 2 mL microcentrifuge tubes and frozen at −20 °C until processing. A sterile head of each swab type was placed into ES, exposed to air briefly (30 s), and placed into the ES to serve as negative controls.

## Sample preparation and DNA extraction

Prior to extraction, 500 μL ES was added to each tube containing a swab head. Tubes were heated to 95 °C for 10 min to lyse cells. A total of 500 μL of 3% bovine serum albumin (BSA) was then added to the tubes to stabilize the reaction. This product was mixed, spun down, and the supernatant was used as template for subsequent PCR reactions (*Bonito et al., 2017*). To clean and surface sanitize leaves, collected leaf materials were first placed in a solution composed of 10% bleach (0.6% active sodium hypochlorite) and 0.1% Tween 20 and agitated for 7 min, followed by rinsing in sterile water and drying with sterile filter paper. Surface-sanitized leaves were placed in wax paper bags and lyophilized. Leaf litter was also lyophilized. Leaf litter was not treated to the same epiphyte/endophyte sampling given its non-living and fragile state. Leaves and leaf litter were respectively pooled by host species and site. These composite samples were ground with 6 mm ceramic beads in 50 mL centrifuge tubes for 5–10 min using a modified paint shaker

(DC-1-C, Miracle Paint Rejuvenator Co, Grove Heights, MN; modified by the Michigan State University Physics Shop).

DNA was extracted from lyophilized plant tissue using the Mag-Bind® Plant DNA Kit (Omega Bio-tek, Norcross, GA, USA), with the recommended ~15 mg of dry tissue. Soil samples were dried with silica gel beads and homogenized, then DNA was extracted from ~0.5 g of processed soils with the PowerMag® Soil DNA Isolation Kit (Qiagen, Carlsbad, CA, USA) following manufacturer's recommendations.

Fungal amplicon libraries were generated with ITS1f-ITS4 primers (*White et al., 1990*; *Gardes & Bruns, 1993*) and DreamTaq Green DNA Polymerase (Thermo Fisher Scientific, Waltham, MA, USA) following previously described protocols (*Lundberg et al., 2013*; *Chen et al., 2018*). Six PCR no-template negative controls were included in library preparation and sequencing. PCR products were visualized under UV light on an ethidium bromide-stained 0.9% agarose gel after separation by electrophoresis. DNA concentrations of samples were normalized with a SequalPrep™ Normalization Plate Kit (Thermo Fisher Scientific, Waltham, MA, USA) and samples were then pooled into a single library. Amplicons were then concentrated 20:1 with Amicon® Ultra 0.5 mL 50K filters (EMDmillipore, Darmstadt, Germany) and purified with Agencourt AMPure XP magnetic beads (Beckman Coulter, Brea, CA, USA). A synthetic mock community with 12 taxa and 4 negative (no DNA added) controls were included to assess sequencing quality (*Palmer et al., 2018*). Amplicons were sequenced on an Illumina MiSeq analyzer using the v3 600 cycles kit (Illumina, San Diego, CA, USA).

## Sequence analysis

Read quality was assessed with FastQC (*Andrews, 2010*). Due to the low quality of generated reverse reads, only forward reads were carried through for further analysis. Sequences were then demultiplexed in QIIME (*Caporaso et al., 2010*). Primers were trimmed with Cutadapt 1.18 (*Martin, 2011*), and the conserved regions trimmed with usearch -fastq_filter (*Edgar, 2010*). Filtered and trimmed reads were then clustered into operational taxonomic units (OTUs) based on 97% sequence similarity with the uparse pipeline (*Edgar, 2013*). Fungal OTU classification was determined with the CONSTAX2 consensus technique comparing RDP Classifier, BLAST, and SINTAX algorithms trained on the UNITE fungal general release dataset from Feb 04, 2020 (*Abarenkov et al., 2020*) with 80% confidence threshold and recommended settings (*Liber, Bonito & Benucci, 2021*). Statistical analyses and plots were prepared in R version 3.6.1 (*R Core Team, 2019*) using the following packages for plotting and data handling: ggplot2 3.2.1 (*Wickham, 2016*), patchwork 1.0.0 (*Pedersen, 2019*), tidyr 1.0.2 (*Wickham & Henry, 2020*), dplyr 0.8.3 (*Wickham et al., 2019*), purrr 0.3.2 (*Henry & Wickham, 2019*), ggpubr 0.2.3 (*Kassambara, 2019*), gplots 3.0.3 (*Warnes et al., 2020*).

Further identification of indicator OTUs that were poorly classified was completed based upon BLASTn searches (*Altschul et al., 1990*) against the Fungal RefSeq ITS nucleotide database with default search settings. Taxa were classified based the following prioritized rules: (1) If 1 taxa was greater than 99.5% identity and 100% query cover, the species rank was assigned, (2) if multiple taxa were greater than 95% identity and

100% query cover, the lowest rank in common was assigned, (3) if at least five taxa had greater than 80% identity and 100% query cover, the lowest rank in common was assigned, (4) if fewer than five taxa with 80% identity and 100% query cover, no taxa was assigned.

## Ecological analysis

A rarefaction curve (rarecurve function in vegan 2.5-6) (*Oksanen et al., 2019*) and OTU table were generated by rarefying to a minimum acceptable depth (function rrarefy from vegan). Within-sample (alpha) diversity was estimated with phyloseq 1.26.0 (*McMurdie & Holmes, 2013*) with both *Shannon (1948)* and *Simpson (1949)* diversity indices. Both indices are shown to be minimally biased, but the Shannon index weights rare species more heavily compared to the Simpson index (*Mouillot & Leprêtre, 1999*). Effects of substrate, host species, and site on within-sample diversity were examined with linear models, following statistical (*Royston, 1995*; Shapiro-Wilk test, shapiro.test function) and graphical checks for normality and transformation to obtain normality. Model fits were assessed for homoscedasticity with Breush–Pagan tests using the function bptest in lmtest 0.9–38 (*Zeileis & Hothorn, 2002*). Site was used as a variable to account for variation due to host genotype, edaphic effects, and microenvironment not captured by substrate or host species effects. The fit of these models was compared with AICctab from bbmle 1.0.23.1 (*Anderson & Burnham, 2002*; *Bolker & R Development Core Team, 2020*). Fungal community differences within and between substrates were visualized based on Bray–Curtis dissimilarities (*Bray & Curtis, 1957*). A Venn diagram and a table of shared OTUs were constructed in VennDiagram 1.6.20 (*Chen, 2018*) with OTUs having at least 0.01% weighted abundance. Indicator taxa analysis was performed with rarefied OTU counts with the multipatt function in indicspecies 1.7.6 (*De Cáceres & Legendre, 2009*).

Ordinations were performed first with PCoA (function cmdscale in vegan) with Bray–Curtis dissimilarity, Sørensen–Dice similarity (*Dice, 1945*; *Sørensen, 1948*), and Jaccard similarity (*Jaccard, 1901*). Based on variance explained, Bray–Curtis dissimilarity was used for subsequent ordination and statistical tests. We applied non-metric multidimensional scaling (NMDS) ordination (metaMDS in vegan) for graphical display of clustering, as clustering was more interpretable than PCoA and yet remained in agreement with later statistical tests. Substrate community compositions were compared by PERMANOVA, for centroids (adonis2 function in vegan for full model, pairwise.perm. manova in RVAideMemoire 0.9–75 for pairwise comparisons) (*Hervé, 2020*), and by PERMDISP, for dispersion (betadisper function in vegan).

## Swab material comparisons

To compare the communities sampled by each swab material, PERMANOVA and PERMDISP were performed as described above for substrate, host species, and site comparisons. Swab materials were further compared by within-sample diversity and the number of reads sequenced per sample. The number of reads sequenced were modeled with a negative binomial mixed model with "glmer.nb" from package lme4 1.1-21

(*Bates et al., 2015*), with swab material as a fixed effect and leaf sampled as a random intercept effect. Within-sample diversity was assessed using a linear mixed model with the same fixed and random effects, with transformations of diversity values to obtain normality. As before, normality was assessed graphically and with Shapiro–Wilk tests, then model fits were assessed for homoscedasticity with Levene's tests (*Fox & Weisberg, 2019*).

## RESULTS

### Samples and sequencing

In this study, a total of 50 samples were collected for fungal community analysis. These were derived from each of four substrates: leaf surface (20), leaf tissue (10), leaf litter (10), or soil (10), referred to hereafter as "epiphyte", "endophyte", "litter", and "soil". From these samples, amplicon sequencing generated 8,184,524 forward reads, which were reduced to 5,859,924 after quality filtering and trimming.

Clustering OTUs at 97% sequence similarity resulted in 3,402 OTUs. Any OTUs for which more than half of reads occurred in negative controls were labelled as possible cross-contamination and excluded from further analysis, leaving 3,366 OTUs in our sample matrix. A rarefaction curve was performed (Fig. S2), which informed the rarefaction to 5,926 reads per sample. After sample selection and rarefaction, 18 epiphyte, 10 endophyte, 8 leaf litter, and 10 soil samples remained.

### Comparison of substrates, host species, and site

Rarefied OTU counts were ordinated through non-metric multidimensional scaling (NMDS) based upon Bray–Curtis dissimilarity. The first two axes of ordination with PCoA based on Bray–Curtis dissimilarity explained the most variance (34.6%), compared to abundance-based Sørensen–Dice (32.6%) or incidence-based Jaccard (25.7%) indices. Dispersion was significant between all substrates (PERMDISP, $p < 0.0001$) and pairwise (Tukey's HSD test, $p < 0.05$) between all pairs of substrates except Epiphyte-Endophyte ($p = 0.99$), and Soil-Litter ($p = 1.00$).

Centroids, compared with Bray–Curtis dissimilarity between rarefied OTU counts, were significantly different between all substrates ($p < 0.001$) and pairwise ($p < 0.005$) comparisons based upon PERMANOVA. When host species and substrate combinations were compared with pairwise PERMANOVA, soil was the only substrate that showed no host effect ($p = 0.95$, all others $p < 0.05$, Table S2). Substrate was the largest contributing variable to community composition ($R^2 = 0.308$, $p < 0.001$), while host species ($R^2 = 0.075$, $p < 0.001$) and site ($R^2 = 0.086$, $p < 0.001$) were also significant but contributed less (Table 1). All interactions were significant ($p < 0.001$) with substrate × site ($R^2 = 0.206$), substrate × host species × site ($R^2 = 0.122$), and substrate × host species ($R^2 = 0.106$) accounting for a substantial part of the variability. Within-sample (alpha) diversity values were transformed and tested for normality (Shapiro–Wilk, $p > 0.10$), and homoscedasticity (Breusch–Pagan, $p > 0.10$). Mean within-sample diversity was significantly different by substrate (ANOVA, df = 3, $p < 0.01$) and by host species (df = 1, $p < 0.05$) for both Shannon and Inverse Simpson indices, but notably not by site (df = 4, $p > 0.10$) (Fig. S3, Table S3).

**Table 1 PERMANOVA tables of substrate, host species, site, and interactions effects on fungal communities both all factor and pairwise.** Included are both the full model comparison and *p*-values from permuted pairwise tests.

**All-factor PERMANOVA test statistics**

| Factor | $R^2$ | F | Pr (>F) |
|---|---|---|---|
| Substrate | 0.3079 | 19.62 | 0.001 |
| Host species | 0.0751 | 14.37 | 0.001 |
| Site | 0.0864 | 4.13 | 0.001 |
| Substrate:Host species | 0.1057 | 6.74 | 0.001 |
| Substrate:Site | 0.2063 | 3.29 | 0.001 |
| Host species:Site | 0.0551 | 2.63 | 0.002 |
| Substrate:Host species:Site | 0.1216 | 2.32 | 0.001 |
| Residual | 0.0418 | | |
| Total | 1 | | |

**Pairwise PERMANOVA comparisons by substrate**

| | Epiphytes | Litter | Soil |
|---|---|---|---|
| Endophytes | 0.0036 | 0.004 | 0.0015 |
| Epiphytes | – | 0.0015 | 0.0015 |
| Litter | – | – | 0.0015 |

Indicator taxa were determined for each of the substrates and substrate groupings, which shows that several leaf-associated taxa were unique to pre-senescent leaves and litter (Table 2). Several indicator taxa are significantly associated with phyllosphere and litter communities, occurring at high abundance across endophyte, epiphyte, and litter (Fig. 1). Several taxa are also significantly associated with substrate-host combinations (Table S4). A Pearson correlation heatmap illustrates similarity between phyllosphere and some litter communities at the OTU-level (Fig. S4).

A total of 657 OTUs were identified as having individual weighted abundances greater than 0.01%. The soil environment had the highest number of unique OTUs (212) and the lowest number of shared OTUs (152). The litter had the highest total number of OTUs (421) and the highest number of shared OTUs (357), although not the highest proportion of shared OTUs (Fig. 2, Table 3). Community composition based upon the 30 most abundant genera across all samples is consistent with expectations of substrate and host species affinity (Fig. 3A). *Archaeorhizomycetes*, *Tuber*, *Inocybe* and other root-associated or ectomycorrhizal fungi (*Tedersoo, May & Smith, 2010*; *Menkis et al., 2014*) were limited to soil samples, while *Ramularia* and *Ampelomyces*, a leaf pathogen and leaf-occuring hyperparasite, respectively (*Falk et al., 1995*; *Videira et al., 2016*) were common in the phyllosphere.

Within each substrate, effects of host species were notable. Endophyte communities of hickory were enriched for *Sphaerulina* spp. (5.13% *vs* 0.030% in maple) while depleted for *Phyllosticta* spp. (0.011% *vs* 5.30%) and *Seimatosporium* spp. (0.045% *vs* 1.88%). Epiphyte communities of hickory were uniquely characterized by *Erysiphe* (11.4% *vs*

**Table 2 Substrate and cross-substrate indicator OTUs.** Indicator OTUs were determined using the multipatt function the indicspecies package. Substrate groupings with no significant indicator OTUs are not displayed, and only those with a false discovery rate (FDR) < 0.05 are included (maximum of five each). Classification against the UNITE with and Fungal RefSeq ITS databases with CONSTAX2 and BLASTn determined the closest known taxa.

| Substrate | OTU | FDR | CONSTAX2 result | BLAST result |
|---|---|---|---|---|
| Epiphyte | OTU_150 | 0.003 | Fungi sp. | *Ceramothyrium* sp. |
| | OTU_56 | 0.003 | *Golubevia pallescens* | Unknown |
| | OTU_58 | 0.003 | Fungi sp. | Unknown |
| | OTU_108 | 0.003 | Mycosphaerellaceae sp. | *Acrodontium* sp. |
| | OTU_191 | 0.003 | *Taphrina vestergrenii* | Unknown |
| Endophyte | OTU_48 | 0.01 | *Phyllosticta minima* | *Phyllosticta* sp. |
| | OTU_83 | 0.012 | *Seimatosporium* sp. | *Seimatosporium* sp. |
| | OTU_34 | 0.022 | *Rhytisma* sp. | *Rhytismataceae sp.* |
| | OTU_683 | 0.033 | *Zygophiala tardicrescens* | *Schizothyrium* sp. |
| | OTU_1104 | 0.034 | *Hypoxylon carneum* | Unknown |
| Litter | OTU_23 | 0.003 | *Colletotrichum* sp. | *Colletotrichum* sp. |
| | OTU_32 | 0.003 | Xylariales sp. | Xylariales sp. |
| | OTU_40 | 0.003 | *Codinaea lambertiae* | *Codinaea lambertiae* |
| | OTU_52 | 0.003 | Ascomycota sp. | *Coleophoma* sp. |
| | OTU_112 | 0.003 | *Lophiostoma* sp. | *Amorocoelophoma* sp. |
| Soil | OTU_19 | 0.003 | *Saitozyma podzolica* | *Saitozyma* sp. |
| | OTU_41 | 0.003 | *Archaeorhizomyces* sp. | *Archaeorhizomyces* sp. |
| | OTU_85 | 0.003 | *Solicoccozyma terricola* | *Solicocozyma terricola* |
| | OTU_93 | 0.003 | *Trichoderma hamatum* | *Trichoderma* sp. |
| | OTU_785 | 0.003 | *Pseudogymnoascus roseus* | *Pseudogymnoascus* sp. |
| Epiphyte + Endophyte | OTU_189 | 0.003 | *Ramularia nyssicola* | *Ramularia nyssicola* |
| | OTU_479 | 0.003 | *Ramularia* sp. | *Ramularia* sp. |
| | OTU_251 | 0.007 | Ascomycota sp. | *Ramularia* sp. |
| | OTU_3326 | 0.009 | *Ramularia* sp. | *Ramularia* sp. |
| | OTU_2 | 0.012 | *Ampelomyces* sp. | Unknown |
| Epiphyte + Litter | OTU_66 | 0.003 | *Bulleribasidium* sp. | *Bulleribasidium* sp. |
| | OTU_68 | 0.003 | *Epicoccum* sp. | *Epicoccum* sp. |
| | OTU_82 | 0.003 | Capnodiales sp. | Dothideomycetes sp. |
| | OTU_165 | 0.003 | *Dioszegia athyri* | *Dioszegia* sp. |
| | OTU_226 | 0.003 | Agaricales sp. | Unknown |
| Endophyte + Litter | OTU_87 | 0.033 | Ascomycota sp. | *Paraconiothyrium* sp. |
| | OTU_1 | 0.033 | Glomerellaceae sp. | *Colletotrichum* sp. |
| | OTU_127 | 0.035 | Diaporthales sp. | *Diaporthe* sp. |
| Litter + Soil | OTU_380 | 0.003 | Ascomycota sp. | Helotiales sp. |
| | OTU_282 | 0.013 | *Striatibotrys eucylindrospora* | *Striatibotrys* sp. |
| | OTU_193 | 0.02 | *Cylindrocladium peruvianum* | *Cylindrocladiella* sp. |
| | OTU_121 | 0.023 | *Chalara* sp. | Leotiomycetes sp. |

(Continued)

| Table 2 (continued) | | | | |
|---|---|---|---|---|
| Substrate | OTU | FDR | CONSTAX2 result | BLAST result |
| Epiphyte + Endophyte + Litter | OTU_4 | 0.003 | *Ramularia* sp. | *Ramularia* sp. |
| | OTU_5 | 0.003 | *Ramularia pratensis* | *Ramularia* sp. |
| | OTU_8 | 0.003 | Dothideomycetes sp. | *Cladosporium* sp. |
| | OTU_10 | 0.003 | Didymellaceae sp. | Didymellaceae sp. |
| | OTU_11 | 0.003 | Pleosporales sp. | *Alternaria* sp. |

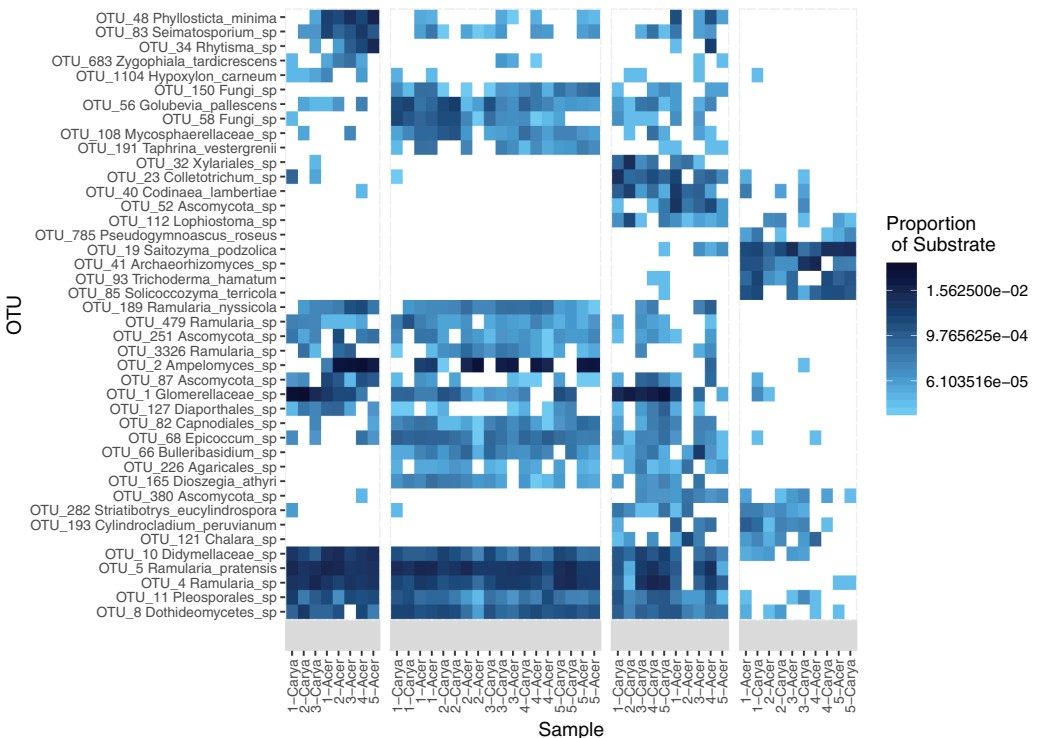

**Figure 1 Heatmap of OTU abundance by substrate.** A total of 42 OTUs were included in the heatmap, all which were significant indicator taxa as determined using the multipatt function in the indicspecies package. Taxa names were determined using the CONSTAX2 classifier. Sample names are included beneath each column, with host species as either *Acer rubrum* (Acer) and *Carya ovata* (Carya) and site number designated in the name.

0.019%) and *Golubevia* spp. (4.64% *vs* 0.70%), with a much lower abundance of *Ampelomyces* spp. (4.54% *vs* 38.9%). OTUs within these genera were significant indicator taxa for their substrate and host species. In leaf litter, some genera were much more abundant in maple litter, including *Lareunionomyces* (4.31% *vs*. 0.010%), *Dothiora* (0.84% *vs* 0.10%), *Ampelomyces* (0.14% *vs* 0.030%), and *Saitozyma* (0.047% *vs* 0.0034%), or in hickory such as *Periconia* (2.14% *vs* 0.051%), *Plectosphaerella* (3.73% *vs* 0.10%), and *Mycosphaerella* (0.40% *vs* 0.030%). *Ramularia* (3.99% in maple *vs* 10.97% in hickory), *Taphrina* (0.061% *vs* 0.14%) and *Scleroramularia* (0.020% *vs* 0.0034%) were between 2- and 6-fold different between host species.

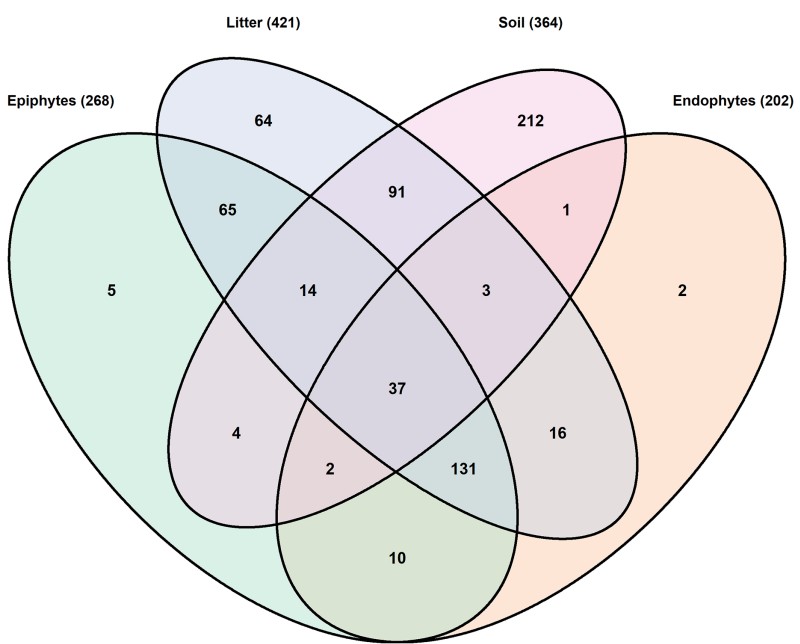

**Figure 2 Venn diagram of shared OTUs by substrate.** A total of 657 OTUs were included which had substrate-weighted abundance greater than 0.01%. Overlapping regions indicate OTUs with at least one read in each of the substrates.               

**Table 3 Total, unique, and shared OTUs by substrate.** Comparison of the number of OTUs in each of four substrates. The total number of OTUs, the number of unique OTUs in each substrate and the number of OTUs shared between substrates are shown. The total number of shared OTUs is the number of unique OTUs subtracted from the total number. OTUs were filtered to exclude those with a substrate-weighted abundance of less than 0.01%.

|  | Epiphyte | Endophyte | Litter | Soil |
|---|---|---|---|---|
| **Total OTUs** | 268 | 202 | 421 | 364 |
| **Unique OTUs** | 5 (2%) | 2 (1%) | 64 (15%) | 212 (58%) |
| **Shared with epiphytes** | – | 180 (89%) | 247 (59%) | 57 (16%) |
| **Shared with endophytes** | 180 (67%) | – | 187 (44%) | 43 (12%) |
| **Shared with litter** | 247 (92%) | 187 (93%) | – | 145 (40%) |
| **Total shared** | 263 (98%) | 200 (99%) | 357 (85%) | 152 (42%) |

Despite differences in communities based on host species, some core taxa were observed at similar abundance between host species within a substrate. Endophyte communities had taxa such as *Ramularia* spp. (13.3% in *A. rubrum vs* 16.6% in *C. ovata*), while in epiphyte communities *Ramularia* spp. (21.5% *vs* 26.2%), *Exobasidium* spp. (2.47% *vs* 4.11%), *Taphrina* spp. (1.51% *vs* 2.11%), and *Scleroramularia* spp. (1.30% *vs* 0.93%) had similar relative abundances between hosts. Similarly abundant genera in leaf litter were *Exobasidium* (0.73% in *Acer rubrum vs* 0.50% in *Carya ovata*), *Seimatosporium* (0.091% *vs* 0.084%), *Inocybe* (0.017% in both), *Mortierella* (sensu *lato*; 0.010% in both), and *Russula* (0.0033% in both). Ordination with non-metric multidimensional scaling (NMDS) and

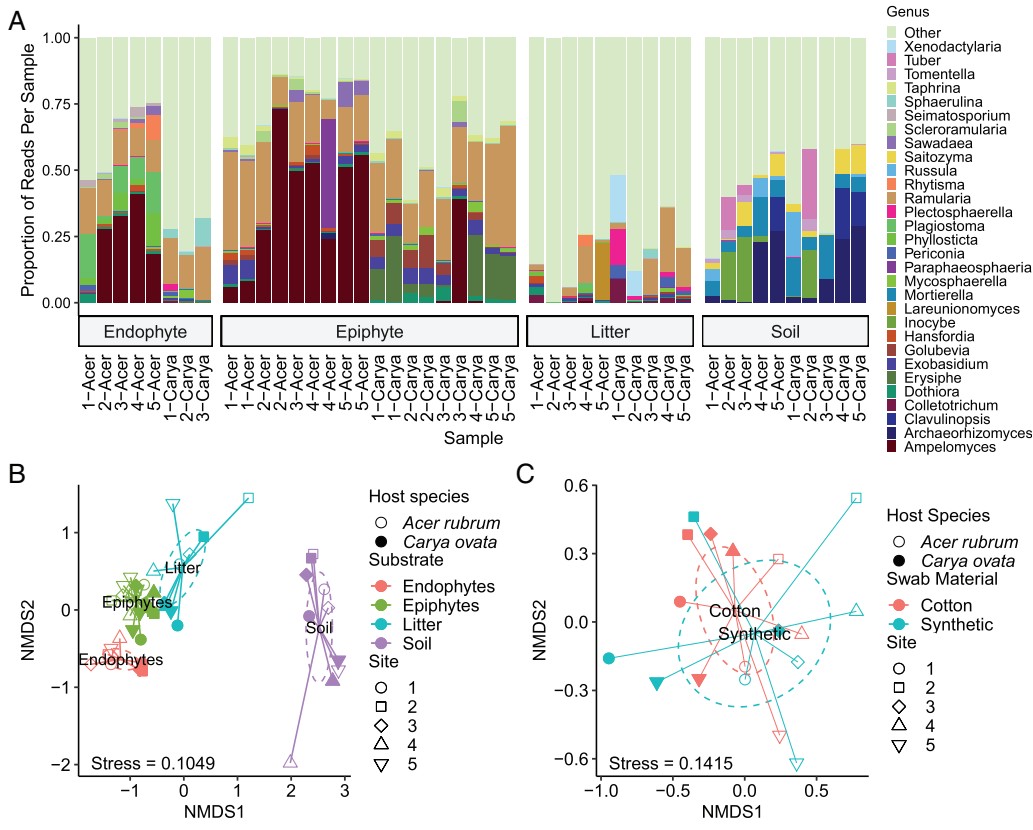

**Figure 3 Comparison of community composition by substrate, host species, site, and swab material.**
(A) The most abundant 30 genera, determined by substrate-weighted abundance and CONSTAX2 classification, are displayed by the proportion of the community composed of OTUs classified within each genus. "Other" includes all OTUs not in the top 30, regardless of abundance. Samples are labeled by host species (genus name) and site with each substrate. (B) Communities were ordinated using non-metric multidimensional scaling (NMDS) and Bray-Curtis distance, with ellipses representing 97% confidence interval estimates of centroids. (C) Epiphyte communities were re-ordinated separately from the remaining samples. Ellipses show 97% confidence interval estimates of centroids of each swab material.

Bray-Curtis dissimilarity show a broad pattern of fungal communities segregating by substrate, host species, and site (Fig. 3B).

## Comparison of swab materials

In this study we compared cotton-tipped and synthetic polymer-tipped swabs for collecting epiphytic samples. Total read counts were modeled using a negative binomial mixed model, with a fixed effect of swab material and a random intercept effect of leaf sampled. Read counts per sample were not significantly different between paired cotton and synthetic swab samples (Fig. S5) (df = 1, $p$ = 0.736, z = −0.338, $n_{samples}$ = 19, $n_{blocks}$ = 10, incidence rate ratio for synthetic: cotton = 0.886, 95% CI [0.432–1.799]). PERMANOVA was used to compare similarity using rarefied OTU counts and Bray-Curtis distance, and showed no significant difference ($p$ = 0.7) between swab types. Dispersion was also not significantly different ($p$ = 0.36). NMDS ordination of swab (epiphyte) samples demonstrated this overlap of centroids and dispersion between

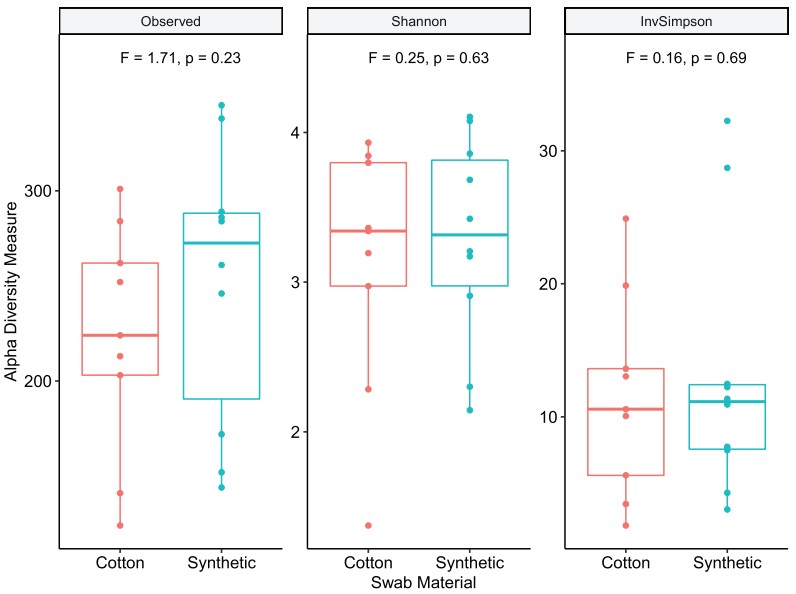

**Figure 4 Richness and within-sample diversity of fungal communities by swab material.** Epiphyte communities sampled with swabs were compared by observed richness, Shannon, and Inverse Simpson estimates of within-sample (alpha) diversity. Diversity estimates were determined using the phyloseq package. Test stastistic F and *p*-values are derived from linear mixed models with swab material as a fixed effect and leaf sampled as a random intercept effect.

material types (Fig. 3C). Comparison of within-sample diversity showed no significant difference between mean diversity for cotton and synthetic samples (Fig. 4), with Shannon diversity (*p* = 0.63, F = 0.24, 95% CI difference [−1.75 to 2.12]) or Inverse Simpson (*p* = 0.69, F = 0.16, CI difference [−0.78 to 1.72]). However, samples collected with the synthetic swab material had somewhat higher diversity for each metric examined.

## DISCUSSION

In this study, we set out to determine how host species, substrate, and site structured fungal communities in a temperate forest ecosystem. We investigated this question with high-throughput ITS amplicon sequencing of fungal communities from pre-senescent leaves, litter, and soils sampled across five sites and between two host species, shagbark hickory (*Carya ovata*) and red maple (*Acer rubrum*). The fungal communities we observed show an effect of each factor characterized. We observed similar dispersion between endophytes and epiphytes, as well as soil and litter, while the litter communities were more similar in composition to pre-senescent leaf communities than to soil communities. Within-sample diversity varied by substrate and host species, but not site. As reported previously, soil fungal communities were more diverse than litter and phyllosphere communities (*Kong et al., 2020*). Our results indicate discrete mechanisms of fungal community assembly and maintenance that differ between substrates, and lead to unique spatially explicit fungal communities based on substrate type. Host species, substrate and site contributed most to community structure in our study. These factors have also been deemed significant in other studies (*Abdelfattah et al., 2019*; *Christian et al., 2020*; *Küngas, Bahram & Põldmaa, 2020*).
## Comparing the phyllospheres of maple and hickory

Phyllosphere microbial communities of *A. rubrum* have previously been examined, but only with culture-based fungal surveys (*U'Ren et al., 2012*) and amplicon-based bacterial surveys (*Laforest-Lapointe, Messier & Kembel, 2016b*). *U'Ren et al. (2012)* isolated endophytic fungi, which were primarily classified as *Pestalotiopsis*, *Phyllosticta*, *Colletotrichum*, *Plagiostoma*, and *Ramularia* spp. These genera accounted for about 29% of reads in our *A. rubrum* endophyte samples. *Seimatosporium* spp. composed an additional 1.88% of these samples. *Ampelomyces* spp. contributed substantially (38.9%) to *A. rubrum* epiphyte communities. While we find consistency across these studies, our culture-independent method allowed for greater sampling depth and likely sequenced taxa that are difficult to culture. *Laforest-Lapointe, Messier & Kembel (2016b)* showed an effect of intra- and inter-individual variability on leaf endophyte communities, which may underlie the variation between sampled hosts we observed.

We are unaware of any phyllosphere leaf fungal community studies in *C. ovata*. Despite growing in the same habitat, our analyses show significant differences in phyllosphere (epiphytic and endophytic) fungal communities between host species, supporting our hypothesis. *Carya ovata* phyllosphere communities were enriched in *Sphaerulina*, *Erysiphe*, and *Golubevia* spp. compared to *A. rubrum*.

## Fungal communities in maple and hickory leaf litter

As leaves senesce and become litter, priority effects and leaf traits are expected to contribute to the communities observed (*Bhatnagar, Peay & Treseder, 2018*; *Veen et al., 2019*). We observed that host species was a significant determinant of phyllosphere communities, and expected that combined priority effects and leaf trait effects would carry forward community differences between the hosts. While most of the taxa detected in leaf litter (421 OTUs, Fig. 2) were also present in the phyllosphere (63%, 266 OTUs) or soil (34%, 145 OTUs), around 15% of the fungal community in the litter was unique to that niche (30% for maple communities, 11% for hickory), of which 17% of the litter unique OTUs were shared between species. Despite a small number of shared OTUs between host species, only a single indicator taxon, annotated as Hypocreales sp. (OTU_50), was found to be significantly associated with hickory litter. Clustering by NMDS and PERMANOVA tests further support differences between host species. Out of the core litter taxa shared between host species, the roles of *Inocybe*, Mortierellaceae, and *Russula* as soil-inhabiting saprobic and ectomycorrhizal fungi (*Geml et al., 2010*; *Seress, Dima & Kovács, 2016*; *Vandepol et al., 2020*) and their low abundance suggests that these taxa may have been detected as the result of soil clinging to sampled litter. However, for other litter core taxa, abundance patterns existing in phyllosphere pre-senescence appear to continue in litter, for example *Exobasidium* spp., or reflect a loss of host specificity in the case of *Seimatosporium* spp.

Fungi detected in the leaf litter are not necessarily derived from either soil or phyllosphere sources. Because the litter we sampled was most likely 1 year old, it may be possible that OTUs are persistent in the litter, or they may turnover or propagate throughout the season or from year to year. Some fungal taxa are known to persist in

temperate forest litter for periods greater than 1 year (*Purahong et al., 2016*). These taxa, such as specialized basidiomycete litter degraders, may be directly colonizing new litter after it settles on the forest floor.

## How do soil and litter fungal communities differ under maple and hickory?

Soils are known to be hyper-diverse environments (*Hu et al., 2019*). In this study, soil fungal communities had the highest number of unique OTUs (364) and lowest proportion of shared OTUs (42%) of any sample type. There were 145 OTUs shared between the litter and the soil, 54 of which were also shared with pre-senescent leaves (19/78 for maple, 37/87 for hickory). These OTUs may have originated in pre-senescent leaves then dispersed to the litter, and then to the soil. Alternately, dispersal could have occurred whereby soil particles carrying fungi reached leaves and litter by wind or rain splash, since soil communities may be reservoirs of phyllosphere diversity (*Zarraonaindia et al., 2015*). However, our study cannot assess the direction of dispersal. Fungal inoculum in rainfall could be another independent source of inoculum in the sampled substrates (*Bell-Dereske & Evans, 2021*). Only 22% (91 of 421) of the OTUs found in the soil were exclusively shared between soil and litter, which indicates a limited contribution of the soil fungal community to that of the litter. Overall, community composition of the soil differed significantly from the other niches sampled (Fig. 3A).

## No differences found between swab types on epiphytic fungal diversity indices

A second research question in this study was whether the swab material used to sample epiphytic communities affected the measured diversity. We found no significant differences between synthetic and cotton swabs in any diversity measurement, including richness, similarity, and read counts. The cost per unit of synthetic swabs are approximately 10 times higher than the cost per unit of cotton swabs, which motivated this research question. These data indicate that future studies can use sterile cotton swabs for sampling without any loss in sampling effectiveness, a more cost-effective strategy.

## Limitations and future directions

Sampling in this study was constrained to a single time point and comparisons of only two host plants. More expansive time-series data would allow for measurements of community assembly, stability, and turnover. *Vořišková & Baldrian (2013)* proposed that some early fungal diversity in leaf litter may carryover from living leaves. It is possible these taxa were missed by the time we collected samples (late summer). Conducting the experiment earlier in the season, along a temporal transect with more time-discrete sampling, or controlling for this effect by using spore traps (*Abdelfattah et al., 2019*) could address this issue, and provide a clearer view of the successional changes in fungal communities during leaf senescence and decay. Further, variability in plant microbiomes exists across scales, from intra-individual (*Osono & Mori, 2004*; *Laforest-Lapointe, Messier & Kembel, 2016b*), to vertically within a canopy (*Izuno et al., 2016*), geographically

within a forest (*Cordier et al., 2012*), between species (*Laforest-Lapointe, Messier & Kembel, 2016a*), and temporally throughout the year (*Materatski et al., 2019*). Our sample design was based on sampling of a single tree of each species at each site, thus, we cannot dissect the specific causes of inter-individual variation because genotype, environment, and other individual factors are confounded. At the individual/vertical scale, we only sampled leaves within 3 m of the ground. Since smaller trees have lower branches, our sampling was biased towards younger trees that were more accessible, and this may have biased the observed phyllosphere communities. Yet, our results provide a significant contribution by identifying drivers of leaf fungal diversity across the leaf-soil continuum.

Our study dissected factors that affect fungal community structure in forest ecosystems. A more extensive sampling approach incorporating metatranscriptomics, amplification-free metagenomics, and/or genome-based accounting of rDNA copy number could reduce the potential biases inherent in our methods, while also providing increased detail of the functional roles of the organisms in the community, distinguishing dormant or commensal organisms from active saprobes, pathogens, hyperparasites, or mutualists. Finally, to relate fungal communities to ecosystem functions and biochemical processes, surveys of community composition should be combined with host plant phenotypes and measurements of nutrients in litter and soil, leading to insights about host fitness and biogeochemical cycling.

## CONCLUSIONS

Our sampling of leaf, leaf litter, and soil microbiomes shows that substrate, host species, site, and the interactions of these factors account for a substantial amount of variation in fungal communities. These factors are indicative of specific influences affecting the community, such as substrate chemistry and local environments. We found that fungal phyllosphere communities differ by host species, an effect which largely persists in the litter community following leaf senescence. We also found that pre-senescent leaf communities overlap substantially with litter community composition, and less so with the soil beneath the leaf litter. Fungal assembly and function of litter communities is complex, thus further studies are needed to address spatial and temporal variability and activity of community members, and their effects on ecosystem processes.

## ACKNOWLEDGEMENTS

We are grateful to Gian M. N. Benucci and Pedro Beschoren da Costa for helpful discussions, analysis recommendations and data processing guidance.

### Funding

This work was supported by the Michigan State University Department of Plant, Soil, and Microbial Sciences (covered sequencing costs), the US National Science Foundation DEB (No. 1737898), the National Institutes of Health NIGMS (No. T32-GM110523), and the

US Department of Agriculture NIFA (No. MICL02416). The funders had no role in study design, data collection and analysis, decision to publish, or preparation of the manuscript.

### Grant Disclosures
The following grant information was disclosed by the authors:
Michigan State University.
US National Science Foundation DEB: 1737898.
National Institutes of Health NIGMS: T32-GM110523.
US Department of Agriculture NIFA: MICL02416.

### Competing Interests
The authors declare that they have no competing interests.

### Author Contributions
- Julian A. Liber conceived and designed the experiments, performed the experiments, analyzed the data, prepared figures and/or tables, authored or reviewed drafts of the paper, and approved the final draft.
- Douglas H. Minier conceived and designed the experiments, performed the experiments, analyzed the data, prepared figures and/or tables, authored or reviewed drafts of the paper, and approved the final draft.
- Anna Stouffer-Hopkins conceived and designed the experiments, performed the experiments, analyzed the data, authored or reviewed drafts of the paper, and approved the final draft.
- Judson Van Wyk conceived and designed the experiments, performed the experiments, analyzed the data, authored or reviewed drafts of the paper, and approved the final draft.
- Reid Longley performed the experiments, authored or reviewed drafts of the paper, and approved the final draft.
- Gregory Bonito conceived and designed the experiments, authored or reviewed drafts of the paper, and approved the final draft.

### Field Study Permissions
The following information was supplied relating to field study approvals (*i.e.*, approving body and any reference numbers):

Samples were taken from Dansville State Game Area, Dansville, Michigan, USA. These public lands are managed by the Michigan Department of Natural Resources and approved by this agency.

### Data Availability
The raw reads are available in NCBI SRA: PRJNA632843.

The summary data, mapping files, and code are available at GitHub: https://github.com/liberjul/Leaf_litter_communities.

## Supplemental Information

Supplemental information for this article can be found online at http://dx.doi.org/10.7717/peerj.12701#supplemental-information.

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
