# Peer review of "Maple and hickory leaf litter fungal communities reflect pre-senescent leaf communities"

_PeerJ, doi:10.7717/peerj.12701_

## Round 0.1 · original submission · Major Revisions

Please address the comments and suggestions of the two reviewers and explain in a clearer way the replications and experimental plan. The text needs a thorough amendment as regards syntax and grammar errors.

Reviewer 1 ·

Basic reporting

The manuscript submitted by Liber and collaborators examines (1) the fungal communities of two forest trees and compare epiphytic-endophytic-litter- soil communities of five different sites within the forest ecosystem, and (2) the effect of different swab material used in sampling on epiphytic community.

The manuscript is clear and well written; it needs editing for the consistency of subtitles formating, use of italics when is needed throughout the text etc.

Experimental design

The manuscript submitted by Liber and collaborators examines (1) the fungal communities of two forest trees and compare epiphytic-endophytic-litter- soil communities of five different sites within the forest ecosystem, and (2) the effect of different swab material used in sampling on epiphytic community. My main concern is related to the experimental approach. As far as I understood, the authors collected samples from four different substrates and five sites in the forest for each plant species only once. This means that the data presented are only a snapshot of the forest fungal communities, thus it is very difficult to conclude on the characteristics, origin etc. of fungal communities detected. Luckily these concerns/limitations are discussed by the authors; still I am not persuaded that research questions stated in the introduction can be really addressed by the presented experimentation.
Concerning the methodology described, it is not clear how many replicates per site and plant are included. Are ten replicates for phyllospheric samples (l 105) and three for the other substrates (l 108) per site included? And then where this “ a total of 50 samples were…. analyzed (l 210)” comes from? Please verify the number of replicates per substrate and site for each plant species. Also, there is no information about the similarity of sample size from the different sites used in the experiment. Please give more details about soil /litter / leaf sampling procedure.

Validity of the findings

No comment

Reviewer 2 ·

Basic reporting

In this study the authors aim to (1) compare the leaf, litter, and soil fungal communities of red maple and shagbark hickory; (2) identify the drivers of fungal community composition in temperate forest substrates; and (3) assess if synthetic and cotton swabs provide similar detection of leaf epiphytic fungal communities. To do so, 5 individuals per tree species were sampled in a forest ecosystem in southern Michigan. For each tree, leaves, litter, and soil were sampled. Swabs and leaf surface sterilization were used to distinguish between leaf fungal epiphytes and endophytes.

The document is professionally structured, the text is mostly clearly written, and the scripts and processed data are available on GitHub.

-Missing references to plant and tree fungal community studies
Lines 57-63 are missing background. I believe discussing the findings of the following references could help provide a sufficient field background.
Kembel, S. W., & Mueller, R. C. (2014). Plant traits and taxonomy drive host associations in tropical phyllosphere fungal communities. Botany, 92(4), 303-311.
Whitaker, B. K., Christian, N., Chai, Q., & Clay, K. (2020). Foliar fungal endophyte community structure is independent of phylogenetic relatedness in an Asteraceae common garden. Ecology and Evolution, 10(24), 13895-13912.
Whitaker, B. K., Reynolds, H. L., & Clay, K. (2018). Foliar fungal endophyte communities are structured by environment but not host ecotype in Panicum virgatum (switchgrass). Ecology, 99(12), 2703-2711.
Zimmerman, N. B., & Vitousek, P. M. (2012). Fungal endophyte communities reflect environmental structuring across a Hawaiian landscape. Proceedings of the National Academy of Sciences, 109(32), 13022-13027.

Experimental design

This manuscript provides interesting results that are well aligned with the aims and scope of PeerJ. The research question / hypotheses are well-defined.

The sampling effort is modest (a total of 50 samples), but could provide sufficient information to make a significant contribution to the literature.

Line 101: “A suitable amount” is not a scientific quantity that can allow fellow researchers to reproduce an experiment. Please state with biologically and scientifically relevant terms what represented a “suitable amount”.

Validity of the findings

One of my main concern with the study design/results/discussion is the confusion between "site" and "individual tree" effects. Indeed, the authors consider their design to allow for detecting a "site" effect but since only one individual per tree was sampled at each of their "site", and since it has been shown that there are inter-individual differences in leaf microbial communities, the results are confounding. For example, the authors could code "site" as a random effect or blocking effect in a linear-mixed model or permanova respectively, where each "site" would include the two individuals from A. rubrum and C. ovata sampled.

Overall, the statistical analyses are sound, but way too much results and discussion space is spent on discussing differences in OTU numbers (lines 246-250, lines 358-365), thus providing very little substance to the field and wasting an opportunity to discuss dispersal mechanisms that could explain the leaf-to-soil continuum.

I believe the authors miss an opportunity to situate clearly in the introduction and discussion how their results contribute to our understanding of the dispersal dynamics across the tree leaf-to-soil continuum. Although very modest, this study provides very interesting results of what happens to fungal communities when they return to the soil.

Additional comments

I commend the authors for using both negative controls and mock communities, for adding a technical aspect to their study (testing swab type differences), and most importantly for openly discussing and acknowledging the limitations of their study.
I suggest for future work that the authors consider moving away from rarefaction as it has been shown to discard considerable amounts of information (McMurdie & Holmes 2014) and there are many demonstrated alternatives implemented in QIIME, QIIME2, and DADA2.

Minor comments:
--Abstract
Line 19: In the first sentence, it would situate adequately the readers by contrasting “living plant microbiomes” with “decaying plant material microbiomes”. It is hard to understand that you mention “living plant microbiomes”, which seemed very strange initially, because your project is going to describe senescent leaf microbiomes too.
Lines 26-28: Please rephrase to “Fungal communities were differentiable based on substrate, host species, and site, as well as all second- and third-level variable interactions.”
Line 28: Ordinations are not used for statistical diagnostic but for visual exploration. Please replace “ordinations” with the appropriate statistical tests employed (i.e. permanovas, permdisp).
Line 29-30: This sentence provides very little meaning. There is dispersal between leaf, litter, and soil. I believe the authors could provide a more insightful final statement and suggest pertinent future avenues.

--Introduction
Line 33: “Plant and soil microbiomes contribute to critical ecosystem functions and are involved in complex interactions within communities”.
Line 34: “Fungi are important microbial community members […]”
Lines 35-39: This sentence is very repetitive with the one at lines 34-35. I suggest taking out the 34-35 and keeping 35-39 but splitting it in two sentences rather than one very long.
Line 58: “Composition” and “structure” are most often used as synonyms for community ecology. Did you want to say: “composition and diversity”?

--Methods
Lines 102-103: “We selected these tree species as they were consistently found across the forest, providing the opportunity to test site effects.”
Line 105: “Ten fresh leaves (Acer) or leaflets (Carya) per tree per site were collected axenically […]”
Line 111: Sterile plastic bags?
Line 213: Why only forward reads? Did the reverse reads fail?
Line 312: “Leaves and leaf litter were respectively pooled by host species and site.” Otherwise it seems like you pooled leaves and leaf litter together.

--Discussion
Lines 327, 366, 374: Should be italicized.
Line 339: Please define what are "the ecologies"?
Lines 392-393: “Yet, our results provide a significant contribution by describing the drivers of leaf fungal diversity across the leaf-soil continuum.”

---

## Round 0.2 · Minor Revisions

Please address the additional comments of Reviewer 2. I believe that the required corrections and additions to your manuscript can be easily addressed but it is important to do so since they will indeed improve the quality of your work.

Reviewer 1 ·

Basic reporting

No comment

Experimental design

No comment

Validity of the findings

No comment

Reviewer 2 ·

Basic reporting

In this second version of their manuscript, the authors have addressed most of the comments from the previous review. Yet, I still have a few important concerns.

Major comment:
In the previous review, I suggested that the authors situate better their study in terms of literature on dispersal mechanisms of microbial communities. They answered: "We decided to not suggest implications for dispersal dynamics because our sampling design does not assess temporal and functional dynamics, and therefore cannot conclude the direction of dispersal or the activity of organisms in each substrate." Yet the two last sentences of their abstract mention dispersal directly. And then part of the discussion adresses dispersal, priority effects, and other drivers of microbial community assembly at lines 320-322 and 347-349. To do that, the authors need to support their statement by looking at important previous works that can support their claims. Their study should be better situated in terms of literature. Like right after their sentence on the distribution of fungal taxa at lines 47-49. The introduction does not provide the needed context that reader needs to understand the discussion.

Experimental design

Lines 114-116: Should mention that there is no opportunity here to look at site effect since it is confounded with inter-individual variation (host genotype). It is better stated later in the methods at lines 200-202 but the authors shouldn't let the reader think that they are able to measure site effect because of their study design. Lines 407-410 should be placed in methods so that the reader can understand the limits of the study design.

Validity of the findings

Hypothesis 82-84 is opposite to literature just cited. Also opposite to literature cited in discussion when results are stated to be in line with lit lines 322-324.

There is no mention that normality and homoscedasticity of data was checked before testing for differences in alpha-diversity. This should be done. If it has been done then mention it at lines 196-197.

Lines 209-212 are confusing since the authors mention using PCoA then NMDS but we only see NMDS in their study. Is it a mistake? Where are the PCoA? Why use PCoA then switch to NMDS?

Lines 240-242: It's not the dissimilarity index that explains variation, it's the two main axes of the ordination on bray-curtis dissimilarities. Rephrase.

Lines 252-254: Where is the test for normality and homoscedasticity?

Line 255: Where is the information on which statistical test was used to identify indicator taxa?

Lines 354-356: Clustering in an ordination is not a statistical test and doesn't support your statement. Remove or rephrase with permanova.

Additional comments

Minor comments:
Line 22: "[...] as well as to the decay [...]"
Line 24: Why does it require further research? Why not state instead that we still ignore most of these factors?
Line 29-31: Last portion of the sentence on second and third level interactions is weird.
Figures: Most are blurry and tough to read.
Figure 4: Add statistical test and significance results.

---

## Round 0.3 · accepted · Accept

Thank you for your submission to PeerJ.